# Renal Protection and Hemodynamic Improvement by Impella^®^ Microaxial Pump in Patients with Cardiogenic Shock

**DOI:** 10.3390/jcm11226817

**Published:** 2022-11-18

**Authors:** Nikolaos Patsalis, Julian Kreutz, Georgios Chatzis, Styliani Syntila, Sebastian Griewing, Carly Pirlet-Grant, Malte Schlegel, Bernhard Schieffer, Birgit Markus

**Affiliations:** Department of Cardiology, Angiology and Intensive Care Medicine, Philipps University Marburg, 35043 Marburg, Germany

**Keywords:** cardiogenic shock, Impella^®^, hemodynamics, renal resistive index, renal organ protection

## Abstract

Acute kidney injury is one of the most frequent and prognostically relevant complications in cardiogenic shock. The purpose of this study was to evaluate the potential effect of the Impella^®^ pump on hemodynamics and renal organ perfusion in patients with myocardial infarction complicating cardiogenic shock. Between January 2020 and February 2022 patients with infarct-related cardiogenic shock supported with the Impella^®^ pump were included in this single-center prospective short-term study. Changes in hemodynamics on different levels of Impella^®^ support were documented with invasive pulmonal arterial catheter. As far as renal function is concerned, renal perfusion was assessed by determining the renal resistive index (RRI) using Doppler sonography. A total of 50 patients were included in the analysis. The increase in the Impella^®^ output by a mean of 1.0 L/min improved the cardiac index (2.7 ± 0.86 to 3.3 ± 1.1 *p* < 0.001) and increased central venous oxygen saturation (62.6 ± 11.8% to 67.4 ± 10.5% *p* < 0.001). On the other side, the systemic vascular resistance (1035 ± 514 N·s/m^5^ to 902 ± 371 N·s/m^5^
*p* = 0.012) and the RRI were significantly reduced (0.736 ± 0.07 to 0.62 ± 0.07 *p* < 0.001). Furthermore, in the overall cohort, a baseline RRI ≥ 0.8 was associated with a higher frequency of renal replacement therapy (71% vs. 39% *p* = 0.04), whereas the consequent reduction of the RRI below 0.7 during Impella^®^ support improved the glomerular filtration rate (GFR) during hospital stay (15 ± 3 days; 53 ± 16 mL/min to 83 ± 16 mL/min *p* = 0.04). Impella^®^ support in patients with cardiogenic shock seems to improve hemodynamics and renal organ perfusion. The RRI, a well-known parameter for the early detection of acute kidney injury, can be directly influenced by the Impella^®^ flow rate. Thus, a targeted control of the RRI by the Impella^®^ pump could mediate renal organ protection.

## 1. Background

Acute kidney injury (AKI) is one of the most common and relevant complications in cardiogenic shock (CS), with prognostic relevance for the patients’ outcomes [1]. The cumulative risk of AKI in CS due to acute myocardial infarction is approximately 20–35%; the 90-day mortality is 12 times higher in patients with AKI than in patients without any impairment of kidney function [2,3,4]. Ultimately, the additional need for renal replacement therapy (RRT) potentiates intrahospital mortality (62% vs. 46%) [5]. Lastly, the risk of developing chronic renal insufficiency, which itself is associated with a significant reduction in quality of life for the patient, shows a 16-fold increase after 5 years [6].

A combination of various pathomechanisms in CS, such as reduced cardiac output, venous congestion, systemic inflammation and the release of vasoactive mediators, is associated with the deterioration of organ perfusion and at least loss of function in AKI [7]. The use of catecholamines in this situation often leads to a pseudo-normalization of the blood pressure values. However, the increase in the systemic peripheral vascular resistance and cardiac afterload due to high doses of catecholamines in turn lead to a further restriction of myocardial function and organ and tissue perfusion. Over the last few years, the use of percutaneous left ventricular assist devices (pLVAD) has become an effective strategy to improve hemodynamics in CS, allowing a reduction of catecholamine doses, ideally also preventing end-organ perfusion [8,9]. Various study results have already shown that acute kidney failure can be avoided, and severe stages of renal insufficiency could be prevented by the timely use of mechanical support devices (e.g., Impella^®^ pump) in CS [10,11,12]. However, little is known about the function and pathomechanism of this effect.

The renal resistive index (RRI) has been traditionally established as an indicator of the occurrence and reversibility of AKI [13,14,15,16]. The RRI is determined by intrarenal artery Doppler measurements (peak systolic velocity minus the end diastolic velocity divided by the peak systolic velocity) and correlates with renal vascular resistance, depicting changes in renal blood flow and microcirculation. The normal range of RRI is about 0.6–0.7 [17]. The RRI is affected by various factors such as age, sex, diabetes, coronary and peripheral artery disease, vascular stiffness and high doses of vasopressor therapy. A high RRI (≥0.7) is associated with an increased risk of AKI and, consequently, high mortality. Patients with RRI values above 0.8 are at an increased risk of RRT and chronic renal insufficiency [15,16,18,19]. According to the authors’ previous data, the Impella^®^ left ventricular assist device seems to positively affect the renal resistive index while increasing laminar blood flow [20]. 

This study aimed to depict the effect on hemodynamics and renal perfusion of the Impella^®^ microaxial pump in patients with myocardial infarction-related CS.

## 2. Methods and Patients

### 2.1. Study Design

This single-center, clinical trial (“RePro-CS”) was conducted over two years (January 2020 to February 2022) at the University Hospital of Marburg, Germany. The trial consisted of 50 patients in CS due to myocardial infarction, supported by the left ventricular Impella^®^ pump (CP).

Inclusion criteria were age more than 18 years, written informed consent, underlying coronary heart disease due to myocardial infarction and CS. Patients with a single kidney, underlying autoimmune or polycystic kidney disease and unstable hemodynamic situations with an expected need for vasopressors or fluids during the scheduled measurements were excluded from the study.

CS was defined as a systolic blood pressure of less than 90 mmHg for more than 30 min or catecholamines required to maintain systolic blood pressure above 90 mmHg. In addition, clinical signs of pulmonary congestion and impaired end-organ perfusion (at least one of the following: altered mental status, cold and wet skin, oliguria with urine output < 30 mL/h or serum lactate > 2.0 mmol/L) had to be present.

In CS patients due to myocardial infarction, Impella^®^ implantation took place within the first 120 min after the initial myocardial event. The Impella^®^ and pulmonary catheter were implanted in the catheterization laboratory prior to the PCI procedure according to established shock algorithms.

After transfer to the intensive care unit (ICU), the first measurement was made 24 h after ICU admission and after having completed the first 24 h of myocardial recovery of left ventricular unloading by Impella^®^. Sixty min before the start of the experimental setting, echocardiography was performed on each patient to verify the correct position of the Impella^®^ pump and to evaluate the stability of ventricular function and sufficient fluid load. This ensured that Impella^®^ support levels could be increased without any suction alert. In addition, invasive hemodynamic parameters had to be stable for the last 6 h before measurements started since neither vasopressors/inotropes nor fluid management were supposed to be changed during this period. According to the authors’ previous data, no relevant changes in systolic, diastolic and mean arterial blood pressure were to be expected while elevating the Impella^®^ support level by 0.44 L ± 0.2 L/min [20].

The first measurement of hemodynamic parameters and RRI occurred 24 h after admission to the ICU. This measurement represents the baseline level of hemodynamic parameters under Impella^®^ support. The second, the safety measurement, took place 60 min later, without any changes in the patient’s therapy. This was to document and ensure that the patient was in a stable hemodynamic situation and that the Impella^®^ level could be changed.

From this point onwards, only the Impella^®^ support level was changed. All other patient and therapy parameters, such as blood pressure and dosages of vasopressors, inotropes and fluids, remained stable and unchanged). Next, the Impella^®^ output was increased by approximately 0.5 L/min. The third measurement of all parameters took place after 30 min on the increased Impella^®^ blood flow when hemodynamic changes and were presumed in steady-state again. After that, the Impella^®^ level was again increased by 0.5 L/min (in total plus 1.0 L/min compared to the baseline measurement). Another 30 min later, the fourth measurement of hemodynamic parameters/RRI were taken from all patients. After that, the Impella^®^ performance level was reduced to reach the baseline level again (minus 1 L/min), and the fifth measurement of hemodynamic parameters/RRI was performed 30 min later. Like the second measurement at the beginning, this was again a safety measurement to document the patient’s hemodynamic stability compared to the initial situation (Figure 1).

### 2.2. Patients

A total of 50 patients with infarct-related refractory cardiogenic shock (CS) supported with Impella^®^ pump were included in this study. The demographics and baseline characteristics of the patients are shown in Table 1.

The mean age of the cohort was 67 ± 13 years, and 74% were male; 36% of the patients had an underlying 3-vessel coronary disease, 26% had a 2-vessel coronary disease, and 28% had a 1-vessel coronary disease. All patients underwent coronary angiography with appropriate revascularization and insertion of Impella^®^ CP through the femoral artery and retrograde through the aortic valve into the left ventricle under fluoroscopic control in the cardiac catheter laboratory. At admission, the mean systolic ejection fraction was 37% ± 18 and mean serum creatinine was significantly elevated at 2.89 ± 2.8 mg/dL at the start of the Impella^®^ support. Mean vasopressor and inotropes dosages were 0.12 ± 0.19 µg/kg/min noradrenaline and 3.1 ± 2.67 µg/kg/min dobutamine during the measurements, representing the difference in the severity of cardiogenic shock in the overall patient cohort.

### 2.3. Renal Resistive Index (RRI)

The RRI was determined by Doppler ultrasound according to standard procedures [21,22]. A transparietal 2–6 MHz pulsed-wave Doppler probe (Philips Sparq, C5-1 ultrasound transducer) was used. Three pulse-wave measurements were conducted on each kidney, and mean RRI values were calculated (6 measurements in total/patient). All RRI measurements were performed by one investigator experienced in kidney Doppler ultrasonography and certified in echocardiography. In order to further assess the intraobserver variability, the RRI was previously measured by the same investigator in a separate cohort of 20 healthy volunteers [20]. The intraclass correlation coefficient (ICC), which was calculated after that measurement, shows a value of 0.997 (95% confidence interval (CI) 0.991–0.999) with a variance of 0.008. According to the literature, the normal range of RRI is between 0.6 and 0.7 [17].

### 2.4. Invasive Hemodynamic Measurement (Pulmonalis Catheter, PAC)

For evaluation of the hemodynamic parameters pulmonal capillary wedge pressure (PCWP), central venous pressure (CVP), central venous oxygen saturation (csvO_2_), cardiac index (CI), cardiac output (CO) and systemic vascular resistance (SVR), an invasive measurement via PA-catheter was conducted.

### 2.5. Clinical Data/Parameters

In addition, various therapy-related data were collected: heart rate; arterial pressure, including systolic, diastolic and mean arterial pressure; catecholamine dosage and volume substitution during the measurements; and respirator parameters in case of invasively ventilated patients. Furthermore, standard laboratory values, including GFR and creatinine at the time point of Impella^®^ insertion, Impella^®^ explantation and discharge and the status of renal replacement therapy (RRT), were documented.

### 2.6. Statistical Analysis

Data are presented as absolute variables and percentages (%) for categorical variables and either median with interquartile range (IQR: 25th–75th percentile) or mean with standard deviation according to the distribution of the variables. After testing for normal distribution using the Shapiro–Wilk and Pearson tests, the paired *t*-test was implemented to test for differences between the various characteristics. Intraobserver variability was calculated based on the ICC and its 95% CI.

All analyses were conducted using SPSS 24 (IBM, New York, NY, USA) and Graphpad Prism 6.0 (GraphPad Software, San Diego, CA, USA). A two-sided *p*-value of less than 0.05 was considered statistically significant.

## 3. Results

It was a prerequisite and should be emphasized again in this context that mean arterial pressure (MAP), heart rate, dosages of catecholamines, fluid management and ventilator settings remained without significant changes during the measurements (Table 2, Figure 2).

The RRI could be routinely calculated for both kidneys in all patients. The mean difference between right and left RRI was 0.02 ± 0.004 *p* = 0.8. No patient had a difference greater than 0.05. After increasing the Impella^®^ support, RRI decreased significantly from baseline 0.736 ± 0.07 to 0.67 ± 0.07 (+0.5 L/min *p* < 0.001) and to 0.62 ± 0.077 (+1 L/min *p* < 0.001) (Figure 3).

Hemodynamic parameters were invasively monitored with a pulmonary catheter according to procedural standards. Enhancing Impella^®^ output by a mean of 0.5 L/min and 1.0 L/min led to a significant increase in the cardiac index from baseline 2.7 ± 0.86 to 3.0 ± 0.98 *p* < 0.001 and to 3.3 ± 1.1 *p* < 0.001 (Table 2, Figure 4), respectively, and of the cardiac output from baseline 5.6 ± 1.9 L/min to 6.1 ± 2 L/min *p* < 0.001 and to 6.6 ± 2.1 L/min *p* < 0.001, respectively. In addition, the central venous oxygen saturation (cvsO_2_) also improved from 62.6 ± 11.8% to 65.1 ± 10.6% *p* = 0.015 and to 67.4 ± 10.5% *p* < 0.001, respectively (Table 2, Figure 3). The systemic vascular resistance (SVR) decreased significantly from 1035 ± 514 s·cm^−5^ to 902 ± 371 s·cm^−5^
*p* = 0.012 after an increase in the Impella^®^ output by 1.0 L/min, whereas increasing the Impella^®^ flow by an average of 0.5 L/min tended to reduce SVR, but the results did not show any significance (1035 ± 514 s·cm^−5^ to 966 ± 392 s·cm^−5^
*p* = 0.1) (Table 2, Figure 4).

RRI values > 0.8 are associated with a significantly increased risk of developing chronic renal insufficiency [15,16]. Therefore, patients were divided into two groups, and 14 patients had a baseline RRI ≥ 0.8. The remaining 36 patients had a baseline RRI < 0.8. In the overall patient cohort, 48% of patients had to undergo an RRT. In this study’s cohort, patients with a baseline RRI ≥ 0.8 had a significantly higher risk of AKI requiring RRT than patients with a baseline RRI < 0.8 (71% vs. 39%, *p* = 0.04) (Figure 5). Both groups had no significant difference in contrast agent exposition during initial intervention (177 ± 77 mL vs. 137 ± 83 mL, *p* = 0.14) (Figure 5). In addition, patients with baseline RRI ≥ 0.8 needed significantly higher doses of noradrenaline and dobutamine than patients with baseline RRI < 0.8 (noradrenaline: 0.36 ± 0.34 µg/kg/min vs. 0.12 ± 0.24 *p* = 0.03; dobutamine: 5.9 ± 2.5 µg/kg/min vs. 3.9 ± 2.5 µg/kg/min *p* = 0.04, respectively) (Table 1). Moreover, the hemodynamic parameters of patients with a baseline RRI ≥ 0.8 show small but not significant changes after enhancing Impella^®^ output by both 0.5 L/min and 1.0 L/min (CI from baseline 2.57 ± 0.53 to 2.74 ± 0.48 *p* = 0.06 and to 2.87 ± 0.6 *p* = 0.07; CO from baseline 5.3 ± 1.4 L/min to 5.81 ± 1.2 L/min *p* = 0.06 and 6.18 ± 1.3 L/min *p* = 0.09; cvsO_2_ from baseline 65.9 ± 10.7% to 64.4 ± 10.5% *p* = 0.3 and to 66.3 ± 10.8% *p* = 0.8). Similarly, SVR could not be significantly decreased (from baseline 955.2 ± 376 s·cm^−5^ to 899.8 ± 251 s·cm^−5^
*p* = 0.4 and 908.6 ± 249 s·cm^−5^
*p* = 0.6). However, in patients with a baseline RRI < 0.8, a significant improvement in hemodynamic parameters after enhancing Impella^®^ output by a mean of 0.5 L/min and 1.0 L/min could be documented. CI raised up from baseline 2.74 ± 1.01 to 3.11 ± 1.1 *p* < 0.001 and 3.38 ± 1.15 *p* < 0.001. CO was elevated from baseline 5.7 ± 2.1 L/min to 6.2 ± 2.2 L/min *p* < 0.001 and 6.74 ± 2.3 L/min *p* < 0.001, respectively. Additionally, cvsO_2_ improved from baseline 62.3 ± 12.4% to 65.6 ± 10.8% *p* = 0.014 and 67.9 ± 10.7% L/min *p* < 0.001. As previously described, no significant alteration of SVR could be seen after increasing Impella^®^ output by a mean of 0.5 L/min (1064 ± 558 s·cm^−5^ to 991 ± 435 s·cm^−5^
*p* = 0.15), whereas an enhancement of support by a mean of 1.0 L/min significantly reduced SVR from baseline 1064 ± 558 s·cm^−5^ to 932 ± 483 s·cm^−5^
*p* = 0.04 (Figure 6).

Interestingly the RRI decreased significantly in both groups. In the group with baseline RRI ≥ 0.8, the RRI dropped from baseline 0.816 ± 0.015 to 0.722 ± 0.056 (*p* = 0.002) and to 0.669 ± 0.08 (*p* < 0.001) after enhancing the Impella^®^ output by a mean of 0.5 L/min and 1.0 L/min, respectively. In the group with a baseline RRI < 0.8 the RRI could also be reduced from baseline 0.707 ± 0.6 to 0.648 ± 0.065 (*p* < 0.001) and ultimately even to low-normal values of 0.614 ± 0.076 (*p* < 0.001) after increasing Impella^®^ output by a mean of 0.5 L/min and 1.0 L/min, respectively (Table 3).

Knowing that RRI values between 0.6 and 0.7 are considered to be in the normal range [17] and that RRI values ≥ 0.7 indicate the onset of AKI [23], we observed that in patients with higher baseline RRI values (≥0.8), the RRI dropped as low as mean to 0.69 during the measurements. However, in the group with baseline RRI values of <0.8, the renal resistive index sank below 0.6, i.e., into an even better, lower normal range (Table 3).

Outside of the original study protocol, in clinical routine, the measurement of the RRI was conducted every 12 h during Impella^®^ therapy. Therefore, we were also able to show that in a short-term follow-up of 15 ± 3 days, patients with a consistent RRI reduction to a value of <0.7 during ongoing Impella^®^ support showed a significant increase in the glomerular filtration rate (GFR) from baseline 53 ± 16 mL/min to 83 ± 16 mL/min (*p* = 0.04) at discharge (n = 8). Patients with a consistent RRI ≥ 0.7 during Impella^®^ support did not show a significant improvement in the GFR (69.9 ± 27 mL/min at baseline to 75.15 ± 31.4 mL/min *p* = 0.5) (Figure 5).

The authors are aware that this additional follow-up data can only be regarded as a tendential indication to evaluate renal function as additional influencing factors cannot be wholly excluded. Patients with ongoing renal replacement therapy are excluded from this data analysis.

## 4. Discussion

Acute kidney injury (AKI) in patients with infarct-related CS is not only associated with increased in-hospital mortality but also with an increased risk of acute and chronic renal failure leading to a considerable decline in the quality of life of the patients. Various study results have already shown that AKI can be avoided, and particularly severe stages of renal insufficiency could be prevented by the earliest possible use of mechanical support devices in CS, e.g., the Impella^®^ pump [10,11,12]. However, little is known about the functional mechanisms of this effect.

This study showed significant changes in systemic hemodynamic parameters and systemic and renal perfusion of patients with infarct-related CS treated with the Impella^®^ pump. Considering the overall cohort, the increase in Impella^®^ output levels by a maximum of about 1.0 L/min up to 2.5 L/min ± 0.4 L/min leads to a significant increase in CO and cvsO_2_, whereas systemic vascular resistance (SVR) significantly dropped. This positive influence on SVR is also reflected in renal perfusion. This is indicated by the data obtained on the changes in the renal resistive index (RRI) during ongoing Impella^®^ therapy. The significant decrease in the RRI after increasing Impella^®^ output, corresponding to the reduction in systemic vascular resistance, and on the other hand, the opposite contemporaneous increase in cvsO_2_, allows conclusions to be drawn about improved kidney perfusion and, ultimately, also protection of organ function.

Our recently published data already showed this effect, where, though in a small patient cohort, even very slight changes in Impella^®^ support level significantly influenced the RRI [20]. At that time, the underlying hemodynamic effects were still unknown.

In contrast to the standard parameters for the evaluation of renal function, such as urine production and the glomerular filtration rate (GFR), the RRI has the advantage of functioning as a very early indicator of the onset of AKI with a prognostic relevance in critically ill patients by reflecting acute hemodynamic changes [13,24].

Normal RRI values range between 0.6 and 0.7 [17], and RRI values greater than 0.7 in critically ill patients indicate the onset of AKI and a higher mortality risk [17]. Moreover, patients in CS with RRI values over 0.8 are more likely to suffer renal failure and require RRT [15,16,18,19]. The results of this study confirm the significance of RRI levels of 0.7 and 0.8. Patients from the study cohort with a baseline RRI greater than 0.8 were significantly more frequently treated with RRT during their hospital stay than patients with a baseline RRI of less than 0.8 during the time on Impella^®^ support. Moreover, hemodynamic parameters of patients with a baseline RRI greater than 0.8 did not improve significantly after increasing Impella^®^ output, whereas hemodynamics of patients with a baseline RRI of less than 0.8 benefitted significantly after augmenting Impella^®^ support. Interestingly, however, the RRI was significantly reduced in both groups after enhancing Impella^®^ output. The question that arises is why the patients with a baseline RRI greater than 0.8 were more likely to require RRT treatment, even though, as a result of the increase in Impella^®^ support, the RRI dropped significantly in both groups. When taking a closer look at the RRI values (Table 3), it became evident that the RRI of the group with a mean baseline value above 0.8 dropped to values of 0.669 ± 0.075 at most, whereas the RRI of the group with a mean baseline value of less than 0.8 plunged to 0.614 ± 0.076 and thus to a very low normal level. Considering as well that patients with a baseline RRI > 0.8 also required significantly higher doses of catecholamines due to the severity of cardiogenic shock and hemodynamic instability, it seems that the functioning of the Impella^®^ alone allows active therapy to prevent the feared complications of CS like loss of organ function. Congruent to this observation are the results of the short-term follow-up of the patients over the period from Impella^®^ insertion to discharge (15 ± 3 days). The GFR improved significantly in patients where a consistent reduction of the RRI to values under 0.7 during Impella^®^ treatment was achieved. On the other hand, RRI values above 0.7 during ongoing Impella^®^ support were associated with a smaller, not significant but nevertheless existent increase in the GFR.

A drop in cardiac output in CS and consecutive heart failure is known to lead to a relevant restriction of renal blood flow, thus aggravating renal venous congestion, which, in addition to a neurohormonal response, including the activation of the sympathetic nervous system, deteriorates renal function independently [25]. The physiological autoregulatory mechanisms of renal circulation decrease renal vascular resistance as a countermeasure to reduced CO to preserve renal perfusion [26]. However, in situations with consistently low CO, such as heart failure in cardiogenic shock, vascular resistance increases due to hyperactivation of the sympathetic nervous system, leading to the deterioration of renal perfusion [26]. In addition, if vasopressors are required in CS, their direct effect on vasoconstriction causes a further decline in renal perfusion while the RRI increases [27]. Percutaneous left ventricular assist devices (pLVAD) such as the Impella^®^ improve hemodynamic parameters in CS, enabling a reduction of catecholamine doses, increased cardiac output and improvement in end-organ perfusion, maintaining organ function [8,9]. The results of this study demonstrate the improvement in hemodynamics and underline the benefit of renal perfusion due to Impella^®^ support. In addition, based on the data from this study, it is fair to assume that achieving the lowest possible RRI values of approximately 0.6 in the context of an individually designed therapy algorithm in affected patients in CS adequately supports kidney perfusion. Thus, the kidney function of patients in CS could benefit significantly and, ideally, be maintained throughout the targeted and RRI-controlled use of the additional Impella^®^ support.

## 5. Conclusions

The results of this single-center study with a short-term design suggest a significant improvement in systemic hemodynamic parameters and renal perfusion via enhanced Impella^®^ support. In addition, short-term observations of this study suggest a significant correlation between lower RRI values during Impella^®^ support and improved renal function, highlighting the RRI as a possible parameter for guided therapy in patients with infarct-related CS. The development of a standardized therapeutic algorithm concerning an individualized Impella^®^ support level based on these data could at least improve the outcome of these patients. Further randomized, controlled studies are now needed to gain further clinical insights into renal function in an acute situation and the clinical courses of patients in CS.

## Figures and Tables

**Figure 1 jcm-11-06817-f001:**
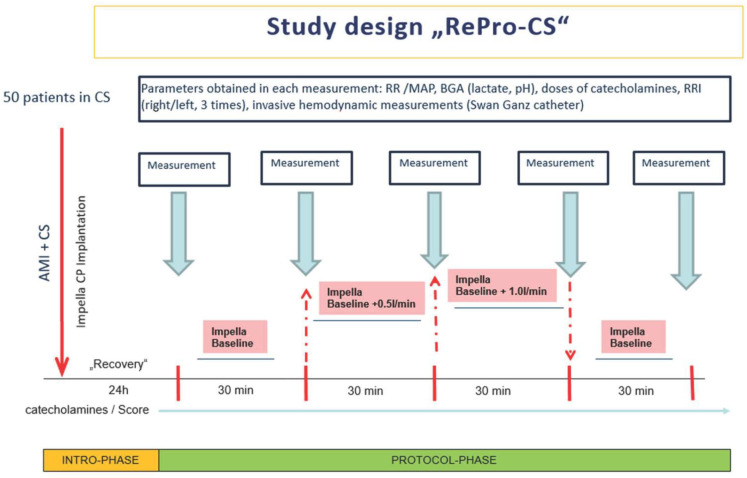
Study design. RR: blood pressure; MAP: mean arterial pressure; BGA: blood gas analysis; RRI: renal resistive index; AMI: acute myocardial infarction; CS: cardiogenic shock.

**Figure 2 jcm-11-06817-f002:**
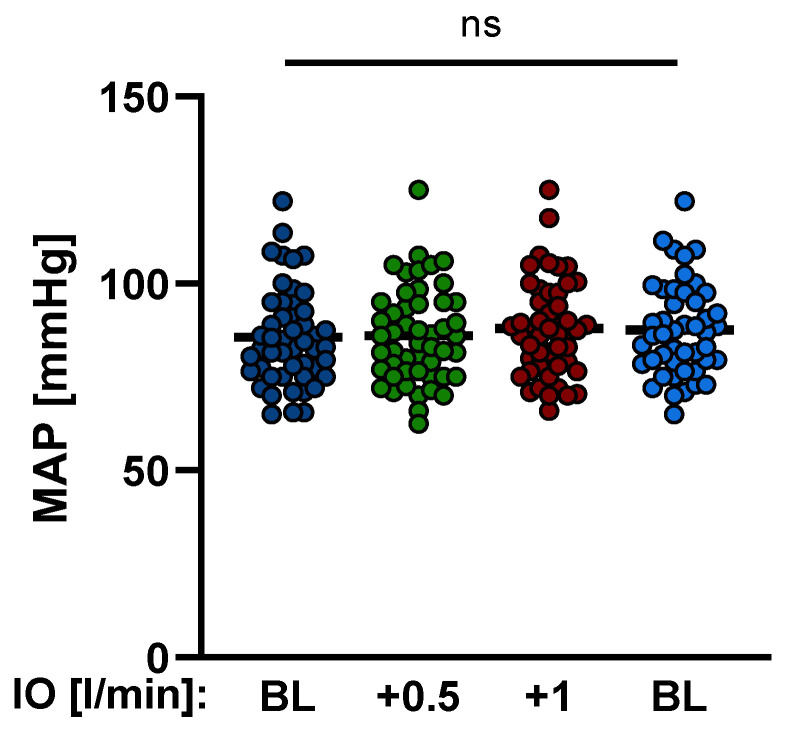
Variation of mean arterial pressure (MAP) during Impella^®^ support. MAP remained without significant change during the measurements. (IO: Impella output; BL: baseline; ns: not significant, *p* ≥ 0.05).

**Figure 3 jcm-11-06817-f003:**
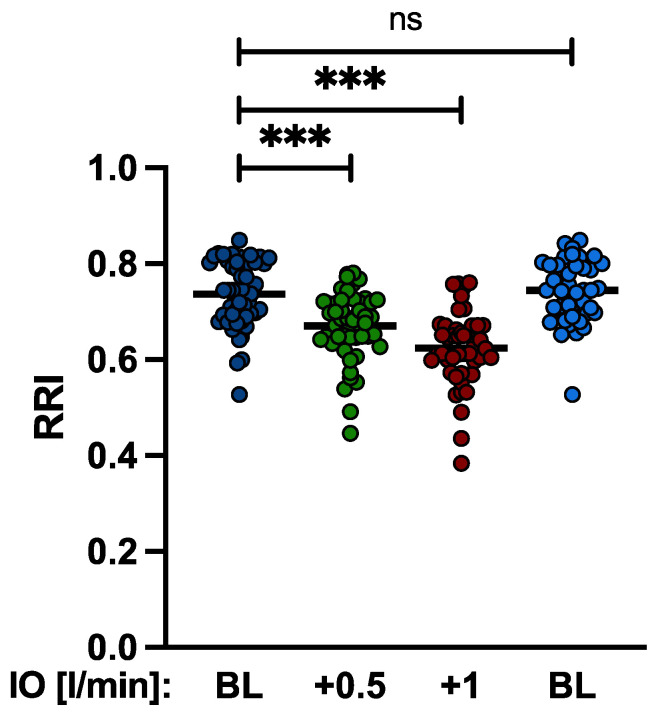
Changes of RRI by Impella output. Increasing the Impella^®^ support by a mean of 0.5 L/min and 1.0 L/min resulted in a significant reduction of the RRI from baseline 0.736 ± 0.07 to 0.67 ± 0.07 and to 0.62 ± 0.077, respectively. (IO: Impella^®^ output; BL: baseline; ns: not significant, *p* ≥ 0.5; ***: *p* < 0.001).

**Figure 4 jcm-11-06817-f004:**
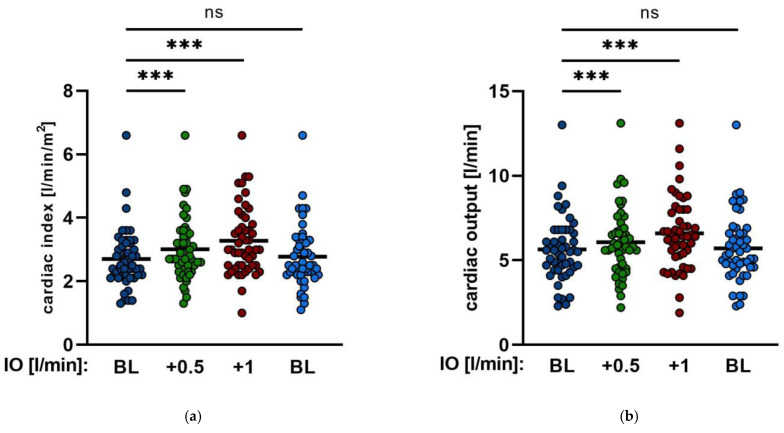
Hemodynamic changes by Impella^®^ output. (**a**,**b**). Cardiac index (CI) and cardiac output (CO) during different changes of Impella support. (**c**,**d**). Central venous oxygen saturation (cvsO_2_) and systemic vascular resistance (SVR) after enhancing Impella^®^ output by mean of 0.5 L/min and mean of 1.0 L/min (IO: Impella output; BL: baseline; ns: not significant, *p* ≥ 0.05; *: *p* < 0.05; ***: *p* < 0.001).

**Figure 5 jcm-11-06817-f005:**
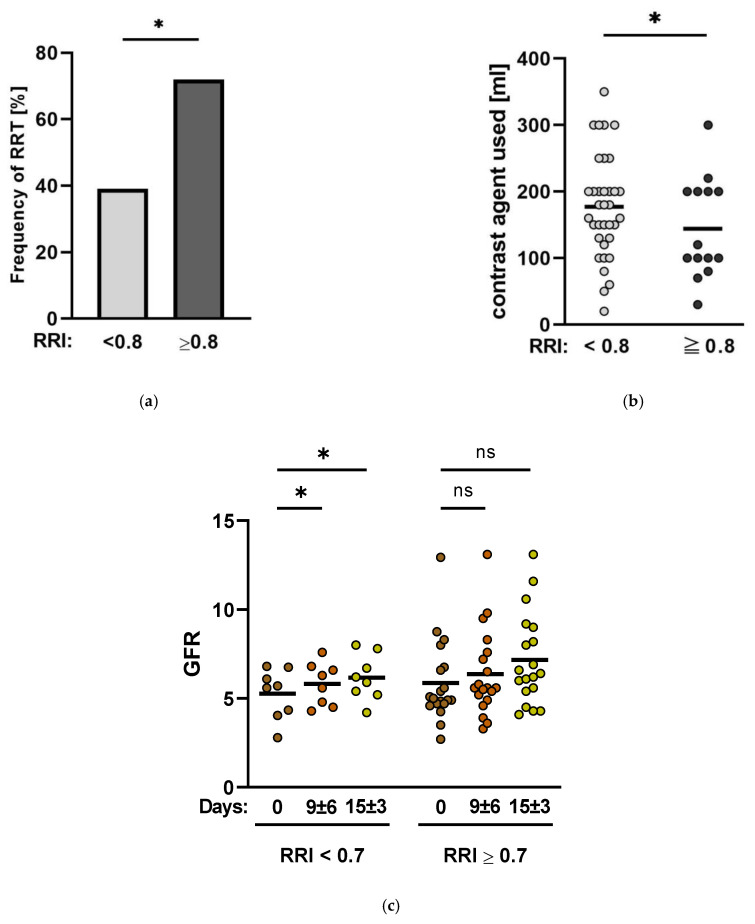
Renal organ function during Impella^®^ therapy. (**a**). Frequency of renal replacement therapy (RRT) by RRI. Patients with a baseline RRI ≥ 0.8 were accompanied by a significantly higher frequency of RRT than patients with a baseline RRI < 0.8 (71% vs. 39% *p* = 0.04). (**b**). Contrast agent exposition during initial intervention. (**c**). Short-term follow-up of GFR over a period of 15 ± 3 days. (GFR: glomerular filtration rate; ns: not significant, *p* ≥ 0.05; *: *p* < 0.05).

**Figure 6 jcm-11-06817-f006:**
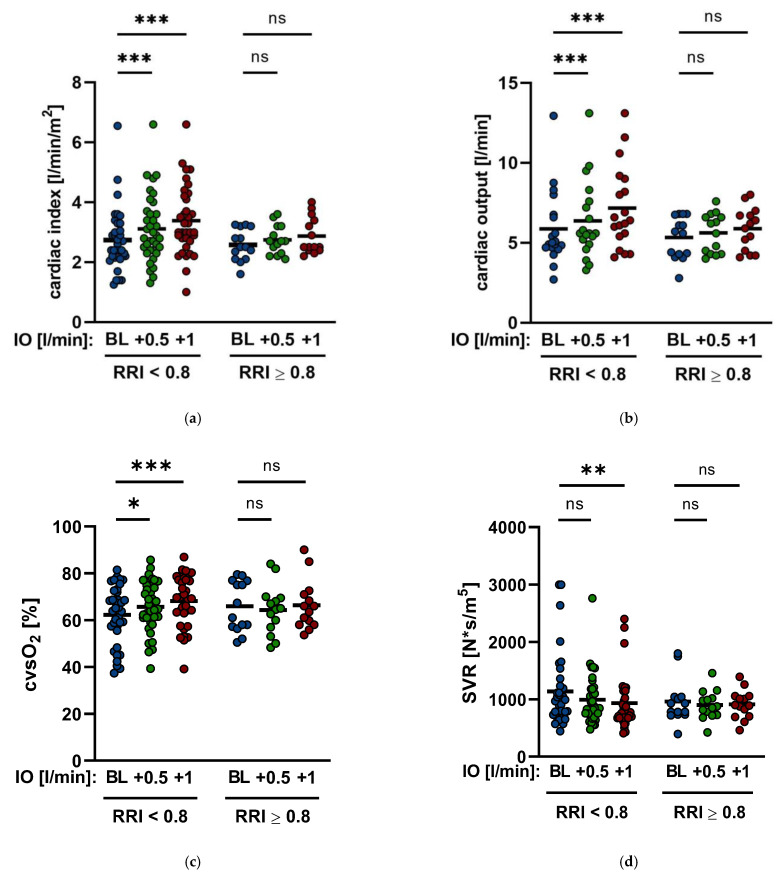
Hemodynamic changes by Impella^®^ output and RRI. (**a**). Cardiac Index (CI) by Impella output (IO) and RRI. (**b**). Cardiac output (CO) by Impella output (IO) and RRI. In patients with baseline RRI ≥ 0.8 CO showed no significant elevation from baseline 5.3 ± 1.4 L/min to 5.81 ± 1.2 L/min *p* = 0.06 and 6.18 ± 1.3 L/min *p* = 0.09, respectively. (**c**). Central venous oxygen saturation (cvsO_2_) by Impella output (IO) and RRI. (**d**): Systemic vascular resistance (SVR) by Impella output and RRI. In patients with baseline RRI < 0.8, SVR could be reduced significantly when increasing Impella^®^ output by 1.0 L/min from baseline 1064 ± 558 s·cm^−5^ to 932 ± 483 s·cm^−5^
*p* = 0.04 (BL: baseline; ns: not significant, *p* ≥ 0.05; *: *p* < 0.05, **: *p* ≤ 0.01, ***: *p* < 0.001).

**Table 1 jcm-11-06817-t001:** Demographics and baseline characteristics. SAP: systolic arterial pressure; DAP: diastolic arterial pressure; MAP: median arterial pressure; LVEF: left ventricular ejection fraction; CHD: coronary heart disease; RRI: renal resistive index.

Demographics and Characteristics	
Age (years)	67 ± 13
Female (%)	26
Male (%)	74
1-vessel CHD (%)	28
2-vessel CHD (%)	26
3-vessel-CHD (%)	36
BMI	26.4 ± 3.4
SAP (mmHg) during measurements	113 ± 19
DAP (mmHg) during measurements	60 ± 11
MAP (mmHg) during measurements	85.4 ± 17
Baseline Creatinine (mg/dL) (at Impella^®^ insertion)	2.89 ± 2.8
Impella^®^ (days)	9 ± 6
Impella^®^-support (min/max. L/min)	1.5/2.5 L/min ± 0.4 L/min
Noradrenaline (µg/kg/min) overall cohort	0.12 ± 0.19
Noradrenaline (µg/kg/min) RRI ≥ 0.8	0.36 ± 0.34
Noradrenaline (µg/kg/min) RRI < 0.8	0.12 ± 0.24
Dobutamine (µg/kg/min) overall cohort	3.1 ± 2.67
Dobutamine (µg/kg/min) RRI ≥ 0.8	5.9 ± 2.5
Dobutamine (µg/kg/min) RRI < 0.8	3.9 ± 2.5
Renal longitudinal length (cm)	9.32 ± 1.1
Renal parenchymal thickness (cm)	2.1 ± 0.4
LVEF at Impella^®^ insertion (%)	37 ± 18

**Table 2 jcm-11-06817-t002:** Parameters of invasive hemodynamic measurement on different levels of Impella^®^ support. MAP: mean arterial pressure; SAP: systolic arterial pressure; DAP: diastolic arterial pressure; RRI: renal resistive index; CI: cardiac index; CO: cardiac output; SVR: systemic vascular resistance; cvsO_2_: central venous oxygen saturation.

	Baseline	+0.5 L/min	+1.0 L/min	Baseline
MAP (mmHg)	85.6 ± 12.1	86.1 ± 12.7	89.1 ± 12.8	87.7 ± 12.5
SAP (mmHg)	110.9 ± 18.1	111.1 ± 18.8	111.53 ± 18.9	111.86 ± 17.5
DAP (mmHg)	59.61 ± 11.1	59.88 ± 11.0	60.22 ± 11.9	60.6 ± 12.4
Noradrenaline (µg/kg/min)	0.12 ± 0.19	0.12 ± 0.19	0.12 ± 0.19	0.12 ± 0.19
Dobutamine (µg/kg/min)	3.1 ± 2.67	3.1 ± 2.67	3.1 ± 2.67	3.1 ± 2.67
RRI	0.736 ± 0.07	0.67 ± 0.07	0.62 ± 0.077	0.74 ± 0.06
CI	2.7 ± 0.86	3.0 ± 0.98	3.3 ± 1.1	2.8 ± 0.98
CO (L/min)	5.6 ± 1.9	6.1 ± 2	6.6 ± 2.1	5.3 ± 2.2
SVR (s·cm^−5^)	1035 ± 514	966 ± 392	902 ± 371	1055 ± 453
cvsO_2_ (%)	62.6 ± 11.8	65.1 ± 10.6	67.4 ± 10.5	64 ± 11

**Table 3 jcm-11-06817-t003:** Extent of reduction of RRI during enhancement of Impella^®^ output by mean of 0.5 L/min and 1.0 L/min in the different patient groups (Baseline RRI ≥ 0.8, <0.8).

RRI	≥0.8 (n = 14)	<0.8 (n = 36)
baseline	0.815 ± 0.12	0.707 ± 0.06
+0.5 L/min	0.722 ± 0.056	0.648 ± 0.065
+1 L/min	0.669 ± 0.075	0.614 ± 0.076

## Data Availability

The authors will submit data supporting the reported results of this study upon request.

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
