# Peer review of "Renal Protection and Hemodynamic Improvement by Impella® Microaxial Pump in Patients with Cardiogenic Shock"

_jcm, 2022, doi:10.3390/jcm11226817_

Round 1

Reviewer 1 Report

This study investigated the effect of the Impella device on renal function in patients with cardiogenic shock.

The abstract gives and adequate summary of the manuscript.

In the introduction the effects of cardiogenic shock on kidney function and subsequently increased rates of renal replacement therapy and mortality are outlined. Drawbacks of catecholamine application are outlined. Previous studies are cited that demonstrated positive effects of the Impella microaxial pump on kidney function but no detailed data on the mechanism are available. The RRI is briefly explained and the aim of the study is formulated.

Methods:

50 patients in cardiogenic shock who were supported by an Impella were investigated. Patients received the device in the acute setting and measurements were carried out after a resting phase on the device.  Additional inotropic support had to be stable. The Impella output was increased by one liter and then reduced again. Hemodynamic and renal parameters were taken during these phases.

The RRI measurement is explained in detail. Invasive hemodynamic measurements were carried out with a Swan Ganz catheter (the term pulmonalis catheter is not used in English).

Statistical methods should be checked. Are methods like ANOVA for repeated measurements necessary for changes in hemodynamic and renal parameters?

Results:

I suggest to move the patient demographics into the patients and methods section. The hemodynamic improvement is well visible in Table 2. Blood pressure remained constant during Impella support whereas RRI decreased during increase of the Impella flow as demonstrated in figures 2 and 3. Results show that a low baseline RRI can result in a more significant drop in RRI with Impella. Data on increase of GFR with low RRI are also shown.

Discussion:

The discussion first states the main findings of the study again and repeats the importance of the cut off value of 0.7  and 0.8 for RRI respectively.   Aspects of the effect of CS on renal blood flow and function are mentioned.

Conclusions are adequately drawn from this single center cohort study. Authors rightfully ask for a prospective randomized trial.

Figures and tables:

In some figures the y axis does show the parameter but not in what units it was measured

Figure legends should be checked for Fig 5

German to English changes:

Hämodynamic should read hemodynamic (see Fig.4)

Reviewer 2 Report

Thank you for submitting you very important work for publication consideration. The study is very interesting, it may provide new information into how a percutaneous left ventricular assist devices (Impella®) could mediate kidney protection in cardiogenic shock associated with myocardial infarction. I have only two suggestions that the authors should review

Methods: Figure 1 Study design  can not be read well. 

References: The bibliographic citations used are more than 5 years old (65.5%)
